# Maresin-1 Prevents Liver Fibrosis by Targeting Nrf2 and NF-κB, Reducing Oxidative Stress and Inflammation

**DOI:** 10.3390/cells10123406

**Published:** 2021-12-03

**Authors:** María José Rodríguez, Matías Sabaj, Gerardo Tolosa, Francisca Herrera Vielma, María José Zúñiga, Daniel R. González, Jessica Zúñiga-Hernández

**Affiliations:** 1Departamento de Ciencias Básicas Biomédicas, Facultad de Ciencias de la Salud, Universidad de Talca, Talca 3460000, Chile; mjrodriguezbecerra@gmail.com (M.J.R.); francisca.herrera@utalca.cl (F.H.V.); maria.zuniga@alu.ucm.cl (M.J.Z.); dagonzalez@utalca.cl (D.R.G.); 2Programa de Doctorado en Ciencias Mención Investigación y Desarrollo de Productos Bioactivos, Instituto de Química de los Recursos Naturales, Universidad de Talca, Talca 3460000, Chile; 3Escuela de Tecnología Médica, Facultad de Ciencias de la Salud, Universidad de Talca, Talca 3460000, Chile; msabaj14@alumnos.utalca.cl (M.S.); gtolosa15@alumnos.utalca.cl (G.T.)

**Keywords:** liver injury, anti-inflammation, growth factors, nuclear receptors, omega-3 derivatives

## Abstract

Liver fibrosis is a complex process characterized by the excessive accumulation of extracellular matrix (ECM) and an alteration in liver architecture, as a result of most types of chronic liver diseases such as cirrhosis, hepatocellular carcinoma (HCC) and liver failure. Maresin-1 (MaR1) is derivative of ω-3 docosahexaenoic acid (DHA), which has been shown to have pro-resolutive and anti-inflammatory effects. We tested the hypothesis that the application of MaR1 could prevent the development of fibrosis in an animal model of chronic hepatic damage. Sprague-Dawley rats were induced with liver fibrosis by injections of diethylnitrosamine (DEN) and treated with or without MaR1 for four weeks. In the MaR1-treated animals, levels of AST and ALT were normalized in comparison with DEN alone, the hepatic architecture was improved, and inflammation and necrotic areas were reduced. Cell proliferation, assessed by the mitotic activity index and the expression of Ki-67, was increased in the MaR1-treated group. MaR1 attenuated liver fibrosis and oxidative stress was induced by DEN. Plasma levels of the pro-inflammatory mediators TNF-α and IL-1β were reduced in MaR1-treated animals, whereas the levels of IL-10, an anti-inflammatory cytokine, increased. Interestingly, MaR1 inhibited the translocation of the p65 subunit of NF-κB, while increasing the activation of Nrf2, a key regulator of the antioxidant response. Finally, MaR1 treatment reduced the levels of the pro-fibrotic mediator TGF-β and its receptor, while normalizing the hepatic levels of IGF-1, a proliferative agent. Taken together, these results suggest that MaR1 improves the parameters of DEN-induced liver fibrosis, activating hepatocyte proliferation and decreasing oxidative stress and inflammation. These results open the possibility of MaR1 as a potential therapeutic agent in fibrosis and other liver pathologies.

## 1. Introduction

Liver fibrosis and the consequent cirrhosis is a worldwide health problem. In 2017, the global burden of chronic liver disease (CLD)-related deaths were more than 2 million [1]. The etiology of CLD can be related to alcoholism, chronic viral hepatitis, autoimmune issues, and nonalcoholic fatty liver disease (NAFLD), among others. NAFLD has a high prevalence in Western countries and it is an important cause of CLD [2]. In CLD, cirrhosis is a terminal stage of damage, where the compensated state progresses to a decompensated disease. Liver fibrogenesis is a dynamic, highly integrated molecular and cellular process, responsible for driving the excessive accumulation of extracellular matrix (ECM) [3]. Moreover, inflammation is the key point in the early stages of fibrosis that can then progress to extensive fibrosis and cirrhosis [4]. Persistent chronic inflammation and an imbalance in oxidative stress may promote liver fibrosis dysfunction, ultimately leading to cirrhosis and its consequences: portal hypertension, hepatocellular carcinoma (HCC) and liver failure [5,6].

Maresin 1 (MaR1) is part of the novel genus of bioactive molecules called specialized pro-resolving molecules (SPM), which are derived from omega-3 fatty acids (eicosapentaenoic [EPA] and docosahexaenoic [DHA] acid) [7,8]. MaR1 is a lipid derivative from DHA and is slightly more potent than other SPMs, such as resolvin D1 (RvD1), in the stimulation of removing dying cells (efferocytosis) by macrophages [9,10]. Moreover, MaR1 accelerates tissue regeneration (in a planaria model), modulates the adaptive immune response by reducing cytokine production acting on the balance between Th1/TH17 cells and tolerogenic T_reg_ cells, and stimulates M1 (inflammatory) to M2 (anti-inflammatory) macrophage phenotype-switches [8,10,11,12]. Currently, there are no reports regarding the role of MaR1 in liver fibrosis, and in general there is scarce information on liver disease, although current data suggest that MaR1 has a hepatoprotective role. Li et al. were the first to describe the potential use of MaR1 on acute liver disease. They found that MaR1 administration mitigated liver damage and had anti-inflammatory and antioxidant effects against acute CCl_4_-induced liver damage [13], associated with MAPKs pathways. After that, Zhang et al. demonstrated that MaR1 reduced the accumulation of inflammatory cells and total liver inflammation in a model of concanavalin-A liver injury [14]. Recently, our group established that MaR1 administration prior to liver ischemia-reperfusion (IR) surgery protected the liver and stimulated hepatocytes proliferation. MaR1 increased the population of restorative macrophages, the movement of Nrf2 from the cytoplasm to the nucleus, and decreased NF-κB at nuclear levels [15]. The hepatoprotective effects of MaR1 on IR liver damage could be related to an activation of the ALXR (lipoxin A4 receptor)/Akt signaling pathway [16]. In non-alcoholic fatty liver disease (NFLD), MaR1 allows hepatocytes to return to homeostasis, reducing apoptosis and increasing phagocytic activity, which is a cardinal sign of active resolution of inflammation [17,18]. Moreover, in a non-alcoholic steatohepatitis model (NASH), MaR1 was found to improve liver damage through the M2 macrophage polarity switch (anti-inflammatory) associated with an increment of retinoic acid-related orphan receptor α (RORα) activity. In that study, the authors found that MaR1 was a ligand of ROR α, and that the MaR1/ROR α/12-LOX autoregulatory circuit would explain the protective effects against obesity and a high-fat diet (Han et al., 2019).

The prospect of liver fibrosis reversibility is striking; data from the histological assessment of biopsies from patients with liver fibrosis, who have successfully been treated, and from animal models of fibrosis, indicate that recovery with the remodeling of the excess of collagen is possible [19,20]. Moreover, the switch from Kupffer cells to an M2 phenotype could be an important piece of liver fibrosis recovery, to be accompanied by hepatocyte regeneration, and ECM degradation [21]. Considering the protective effects of MaR1, and their previously described anti-inflammatory activity, we propose that MaR1 administration has a hepatoprotective effect against fibrosis and could promote the reversibility of liver damage.

## 2. Materials and Methods

### 2.1. Ethics Statement

Experimental animal protocols and procedures complied with the Guide for the Care and Use of Laboratory Animals (National Academy of Sciences, NIH Publication 6-23, revised 74 1985) and were approved by the Bioethics Committee for Care and Research in Animals, Universidad de Talca (Folio 2016-06-B and C).

### 2.2. Model of Fibrosis

Liver fibrosis was induced by an intraperitoneal injection (i.p) of diethylnitrosamine (DEN) (NO258; Sigma -Aldrich, Merck, Darmstadt, Germany) at 70 mg/g body weight per week, for four weeks, according to the model proposed by Kim and adapted by Rodriguez [22,23].

### 2.3. Animals Preparation

Male Sprague-Dawley rats (90–110 grs) were obtained from Bioterio Central, Universidad de Talca. The animals were allowed free access to food (Champion S.A., Santiago, Chile) and water *ad libitum*. The animals were housed in a temperature-controlled room, on a 15 h light/dark cycle. All animals were randomly divided into four groups, as follows: (1) the control group (0.9% NaCl [saline solution], as vehicle DEN + 0.005% ethanol in saline solution as vehicle MaR1); (2) DEN group (DEN + vehicle MaR1); (3) MaR1 group (vehicle DEN + MaR1), and 4) MaR1+ DEN. MaR1 (CAY 10878, Cayman Chemicals, Ann Arbor, MI, USA) was injected i.p twice a week, at doses of 4 ng/g [15]. After four weeks, the animals were fasted (6–8 h) and then anesthetized with 1 mg/Kg Acepromacine Maleate 1% (Pacifor^®^, Drag-Pharma, RM, Chile), 5 mg/Kg Xylazine 2% (Xilagesic^®^, Drag-Pharma) and 50 mg/Kg Ketamine clorhidrate 100 mg/mL (Ketamil^®^, Ilium Laboratories, Troy Animals Health Care, Braddon, NSW, Australia). Liver tissue samples were weighed, frozen in liquid nitrogen, and stored at −80 °C, or fixed in phosphate-buffered formalin and embedded in paraffin for further histology and immunohistochemistry analysis.

### 2.4. Determination of Biochemical Parameters

Alanine aminotransferase (ALT), aspartate aminotransferase (AST), Albumin, alkaline phosphatase (AP), lactate dehydrogenase (LDH), and gamma glutamil-tranferase (γ-GT) were measured using a specific diagnostic kit (ALT, AST and Valtke^®^ Diagnostic Kits, Ñuñoa, Chile; Albumin, AP, LDH and γ-GT by LiquidColor Human™, Wiesbaden, Germany). To control the measurements, adequate two-level controls, normal and pathological, were used. ELISA kits were used for the assessment of the serum levels (pg/mL) of TNF-α (Cat. EH2ILTN, Thermo Sc., Rockford, IL, USA); IL-1 β (Cat. EH2IL1B2, Thermo Sc.); IL-10 (Cat. EH2IL102, Thermo Sc.). All the assays were evaluated in serum and made triplicate. Glutathione (GSH) and glutathione disulfide (GSSG) contents (μMol/g liver) were determined using a Glutathione Assay Kit^®^ (Cat. 703002, Cayman Chemicals), using 400 mg of liver tissue homogenized in cold PBS (PBS pH 6.5 in 1 mM EDTA).

### 2.5. Liver Histology

For hematoxylin and eosin (HE) and Masson’s trichromic stain, the formalin-fixed liver tissues were processed with the automatized tissue processor Leica TP1020 (Leica Microsystems Inc., (Schweiz) AG, Heerbrugg, Switzerland) and Leica EG11504 H (Leica Microsystems Inc.) HE. Masson’s trichromic, and Weigert hematoxylin (for Masson contrast) were acquired in Merck (Darmstadt, Germany). All the liver extracted were at least developed by duplicated and considering 6 fields per slide; the analysis was made by a blind expert Pathologist, following the punctuation of Korourian and Ishak, modified from Goodman [23,24,25,26]. The histological analysis was made in a Leica DM500 microscope (Leica Microsystems) with a high-definition digital camera Leica ICC50W (Leica Microsystems) connected to a LAS EZ software (Leica Application Suite, Heerbrugg, Switzerland).

### 2.6. Immunohistochemistry Studies

After deparaffination and hydration of the liver histological sections, the antigen recuperation was made at 95 °C in citrate buffer (Na_3_C_6_H_5_O_7_ 10 mM, pH 6.0). Endogen peroxidase activity was blocked with a 3 % H_2_O_2_ solution. Unspecific binding was blocked by a bovine serum solution (BSA 3%, sodium azide 15 nM). The primary antibodies used were anti-Ki67 (polyclonal 1:300, Cat. N° PAS-19462, Thermo Fischer Sci., Rockford, IL, USA) and anti-alpha smooth muscle actin (monoclonal, 1:25, Cat N° NCL-L-SMA, Leica Biosystems). The Vectastain^®^ ABC kit (Vector Laboratories Inc., Burlingame, CA, USA) was used for biotinylated secondary antibody assay, and the samples were revealed using ImmPACT^®^ DAB kit (Vector Laboratories Inc.). Harris’ hematoxylin was used as counterstain. The analysis of the slices was made as described in the previous section. The analysis of the positive areas was determined by the Fiji-ImageJ software (NIH, Bethesda, MD, USA).

### 2.7. Western Blot Analysis

Cytoplasmic and nuclear extracts were obtained, adapting the protocol of Deryckere and Gannon [15,27]. Briefly, liver samples (100–500 mg) frozen in liquid nitrogen were homogenized and suspended in a buffer solution pH 7.9, containing 10 mM HEPES, 1 mM EDTA, 0.6% NP-40, 150 mM NaCl, and 0.5 mM PMSF, followed by centrifugation (3020 *g* for 5 min). The supernatant corresponds to cytoplasmic proteins. For nuclear fractions, the pellet was resuspended in a buffer solution 2, containing glycerol 25%, HEPES 20 mM pH 7.9, NaCl 420 mM, MgCl_2_ 1.2 mM, EDTA 0.2 mM, DTT 0.5 mM, PMSF 0.5 mM, benzamidine 2 mM and inhibitor de proteases 80 µg/mL (Pierce Protease Inhibitor Mini Tablets, Thermo Scientific, Rockford, IL, USA). This was followed by centrifugation at 13,000× *g* for 60 s and the supernatant was incubated for 20 min in ice. Then, the supernatant was centrifuged at 13,000× *g* for 30 s at 4 °C, to eliminate nuclear debris (precipitate). Protein fractions (50 µg) were separated in 12% polyacrylamide gels using SDS–PAGE and transferred to nitrocellulose membranes, which were blocked for 1 h at room temperature, with TBS containing 4.5% bovine serum albumin. The blots were washed with TBS containing 0.1% Tween 20, hybridized with rabbit polyclonal primary antibodies, for either: iNOS (1:500, Cat. AB5382, Merck); Cyclin D1 (1:500, Cat. CC1_2_, Sigma-Aldrich, Merck, The Netherlands), p-BCL-2 (1:500, PAS-36742, Invitrogen, Thermo Fisher Sci. Waltham, MA, USA); cleaved caspase 3 (1:100, Cat sc-56052, Santa Cruz Biotech, Dallas, TX, USA); IκBα (1:800, Cat. 07-1483, Merck); p-IκBα (1:800, Cat. sc-8404, Santa Cruz Biotech); Nrf2 (1:500, Cat. sc-722, Santa Cruz Biotech); NF-κBp65 (1:1000, Cat. 06-418, Merck); Keap-1 (1:500, Cat. 8047, Cell Signaling Biotech, Danvers, MA, USA); TGBβ (1:1500, Cat. 3711, Cell Signaling); TGFβ RII (1:250, Cat. sc-17791, Santa Cruz Biotech); IGF-I (1:500, Cat. sc-518040, Santa Cruz Biotech); IGF-I R (1:500, Cat. 3027, Cell Signaling Biotech); GAPDH (1:3000, Cat. 5174, Cell Signaling Biotech); Histone H1 (1:350, Cat. sc-8030, Santa Cruz Biotech), or β-actin (1:3000, Cat. 3700, Cell Signaling biotech). The antibodies were incubated overnight at 4 °C. After extensive washing, the antigen–antibody complexes were detected using horseradish peroxidase-labeled goat, anti-rabbit, IgG/anti-mouse or rabbit (Cell Signaling) and the protein was detected with the kit of protein detection Westar Antares^®^ (Cat. XLS142.0250, Cyanagen, Bolonia, Italy). The chemiluminescent signals were analyzed in the Omega Lum^®^ System (Aplegen, San Francisco, CA, USA), and the quantification of luminescent images was made in ImageJ (NIH). All the original representative images for Western-blot are included in Appendix A.

### 2.8. Statistical Analysis

All data are presented as means ± SD. The number of samples is indicated in each figure. Statistical analyses were performed with GraphPad Prism^®^ software, version 9.2.0 (GraphPad Software, Inc. San Diego, CA, USA). T-student (unpaired data) and one-way ANOVAs were realized. As a post-hoc test, Tukey’s multiple comparison test and Batlest test were used to assess the statistical significance of the differences among the groups. A value of *p* < 0.05 was considered significant.

## 3. Results

### 3.1. MaR1 Administration Decrease Parameters of Chronic Liver Injury

Animal and liver weight was evaluated (Figure 1A–C), and it was found that MaR1 administration promoted a slight weight gain but did not affect the normal development of the liver, considering the four weeks of these assay. Conversely, the DEN administration promoted body weight and increased liver weight by 18% compared with the control (*p* = 0.016), and 21% and 13% compared with MaR1 and MaR1 + Den (*p* = 0.0003 and *p* = 0.0141, respectively), which is an indicator of liver injury.

The levels of AST and ALT were up-regulated in DEN livers (Table 1) and the administration of MaR1 attenuated the increase in these biomarkers (*p* < 0.0001). Moreover, MaR1 administration normalized the albumin levels observed in DEN groups (*p* = 0.007). The levels of γGT, AP and LDH were slightly decreased in the MaR1 + DEN groups, compared with DEN, without statistical significance.

Hepatic fibrosis was evaluated by hepatic histological analysis HE staining (Figure 2A–D) (other stain techniques are shown in Appendix A) and serum biochemical parameters (albumin, AST, ALT, γGT, AP and LDH; Figure 2E). The control and MaR1 + DEN groups showed normal histoarchitecture without necrosis or inflammatory infiltrates, whereas DEN animals showed degenerative changes, with extensive areas of necrosis associated with the loss of cytoarchitecture, inflammatory cell infiltration and fragmented hepatic nuclei, compared with the control and MaR1 groups. The quantification of cytoarchitecture, inflammation and necrosis (Figure 2B–D) showed that the MaR1 administration ameliorated DEN injury by decreasing the degree of architecture distortion, inflammatory infiltration and necrosis areas (3.1, 2.7 and 4.1-fold, respectively, *p* < 0.0001).

### 3.2. MaR1 Reduces Inflammatory Parameters in DEN-Induced Liver Injury

A cytokines analysis for pro-inflammation/anti-inflammation response was determined (Figure 3A–C). The pro-inflammatory cytokines TNF-α and IL-1β were increased in 6 and 5.5-fold, in relation to the control groups (*p* < 0.0001), and 4.8 and 2.4-fold in relation to the MaR1 + DEN group (*p* < 0.0001), with non-statistically significant differences between the control and MaR1 + DEN (0.36 ± 0.42 [control]; 0.53 ± 0.4 [MaR1] and 0.45 ± 0.42 [MaR1 + DEN] for TNF-α and 0.37 ± 0.32 [control]; 0.42 ± 0.29 [MaR1]; 0.84 ± 0.09 [MaR1 + DEN] for IL-1β). On the other hand, the anti-inflammatory IL-10 was measured as showing a slight decrease in the DEN group, compared with the controls (*p* = 0.91); however, MaR1 + DEN presented a statistical increment compared to the control and MaR1 (3.5 and 3.4-fold; *p* < 0.005) and DEN group (5.5-fold *p* < 0.0001).

### 3.3. MaR1 Shows an Anti-Fibrogenic and Proliferative Response against DEN Injury

To evaluate fibrosis, α-SMA and Masson’s Trichrome staining were assayed (Figure 4, panels A and B). An analysis of the extracellular matrix (ECM) deposit revealed, as expected, an exaggeration of EMC deposit revealed by an increase in α-SMA. Masson’s Trichrome Masson positive areas in the DEN animals showed a tendency to form bridges, a situation not observed in the other groups. MaR1 administration reduced the fibrotic areas; specifically, in α-SMA and Masson’s Trichrome fibrosis quantification, there was a 2 and 68.7-fold (respectively, *p* < 0.0001) increase in the DEN group compared with the control. MaR1 administration reduced this value in a 1.4-fold (*p* = 0.0033) and 3.2-fold (*p* = 0.0006), respectively.

To evaluate the proliferative and antiapoptotic responses, the levels of cyclin D1, *p*-BCL-2 and cleaved caspase-3 were measured (Figure 5A–D). Cyclin D1 was increased in the MaR1 + DEN group, in a media of 45-fold (*p* < 0.0001), compared with all other groups. No significant differences were found between the other groups. DEN caused a reduction of *p*-BCL-2 levels in a 1.3-fold (*p* = 0.002), a situation reversed by MaR1 administration, with a 1.4-fold increase (*p* = 0.005), compared with the MaR1 + DEN and DEN groups.

In the case of cleaved caspase-3, DEN-treated animals showed a 3.1, 3.2 and 2.4 increase (*p* < 0.005) compared with the control, MaR1 and MaR1 + DEN groups, with no differences among the other groups.

To evaluate the pro-proliferative activity induced by MaR1 treatment, we measured Ki67 positive areas (Figure 5E), observing an increment in 2.8-fold (*p* < 0.0001) in DEN-treated animals, compared with the control group; however, MaR1 increased even more, in 3.5 and 1.3-fold (*p* < 0.0001), compared with the control and DEN groups, respectively.

### 3.4. MaR1 Protects from Fibrosis Liver Injury through Nuclear Receptors and Growth Factors through Activation Mechanisms

Next, we evaluated the activation of NF-κB and Nrf2 transcription factors as signaling responses of the anti-inflammatory effect of MaR1 treatment. For this, we evaluated the nuclear translocation of nuclear factors NF-κBp65, Nrf2, and the protein expression of related regulator’s transcription activity molecules (Figure 6). As related protein regulates by these transcription factors, it was evaluated by (i) IκB and phospho-IκBα (*p*-IκBα), (ii) iNOS for NF-κBp65 (Figure 6A,D,G,H), and (iii) Keap-1 and (iv) GSH/GSSG tissue levels for Nrf2 (Figure 6A,D,I,J). DEN administration promoted a NF-κBp65 nuclear translocation (3.9-fold, *p* < 0.0001) with a decrease in the cytoplasmic fraction (1.3-fold, *p* = 0.0007), compared with the control group. The MaR1 administration did not modify the NF-κBp65 cytoplasmic (*p* = 0.5942) and nuclear (*p* < 0.0001) content, compared with the control. Moreover, MaR1 + DEN did not modify the *p*-IκBα/IκBα ratio (Figure 6A,G and Appendix A), in contrast with the DEN groups, that showed an enhancement of the *p*-IκBα/IκBα ratio of 3.1, 3.3 and 3-fold, with respect to the control, MaR1 and MaR1 + DEN (*p* < 0.0001). To reinforce the improvement of MaR1 on anti-inflammatory signaling, expression levels of inducible nitric oxide synthase (iNOS) were assessed (Figure 6A,H), finding that the DEN group presented an overexpression of iNOS, compared with the control, MaR1 and MaR1 + DEN groups (2.64; 3,1 and 4.08, respectively *p* < 0.001).

Regarding the nuclear translocation of Nrf2 (Figure 6A,C,D,F), the DEN group showed a decrease in cytoplasmic content of 0.5-fold (*p* < 0.0001), but MaR1 + DEN showed a more important decrease in the levels of Nrf2, by 5.6 and 2.7-fold (*p* < 0.05) to the control and DEN groups, respectively. Conversely, nuclear Nrf2 increased slightly, compared with the control (0.09 ± 0.16 [control] 0.791 ± 0.96 [DEN] *p* = 0.0127) and MaR1 had the most notorious increase in relation to the other groups, with 31-fold (control), 3.8-fold (DEN) and 21-fold (MaR1) (*p* < 0.0001) Nrf2 enhances. Keap-1 protein was enhanced in the DEN group by 5.8-fold (*p* < 0.0001), compared with the control, and MaR1 + DEN by 2.8-fold (*p* < 0.0001), which was not statistically significant when compared with the control. In terms of oxidative stress regulated by Nrf2, the DEN group presented a reduction in GSH of 1.3-fold, compared with normal liver tissue (*p* < 0.0001), associated with an increment of GSSG (1.5-fold *p* < 0.0024) and a fall of the GSH/GSSG ratio (2.3-fold, *p* < 0.0001) (Figure 6H and Appendix A). Moreover, MaR1 reverted the GSH-GSSG dysfunction, normalizing the levels of GSH, GSSG, and the GSH/GSSG ratio (*p* < 0.0001). To ensure that the results were not associated with a loss of components in the assay, we determined the total glutathione content in the liver tissue (2GSH + GSSG) and non-statistical significance was observed between groups *p* = 0.9765) (see Appendix A).

Next, we analyzed growth factors (Figure 7) as possible autocrine/paracrine mediators of fibrosis and proliferation in this model. DEN-treated animals showed an increase in the level of expression of TGF-β and its receptor TGF-β RII (Figure 7A–C) in 1.34 and 2.4-fold compared with the control group (*p* < 0.005); however, it was reduced 1.1-fold (*p* = 0.026) in the MaR1 + DEN group. Moreover, MaR1 + DEN depleted in 2.8-fold (*p* < 0.0001): the increase observed in DEN group, with non-statistical difference among groups.

In the case of insulin growth factor (IGF-I) and its receptor (Figure 7A,D,E), the DEN group showed reduced levels of both molecules in 1.7 and 38-fold (*p* < 0.05), compared with the control group. MaR1 + DEN showed a normalization of IGF-I (*p* < 0.0001) and IGF-IR (*p* = 0.0108), with respect to the enhanced levels observed in DEN animals, and with non-statistical significance among the controls.

## 4. Discussion

When chronic liver damage has been documented, liver fibrosis emerges as an intermediate step between reversion and progression to final organ failure. If fibrosis in CLD can be reversed, the obligatory mechanisms that must be orchestrated include: the switch in the inflammatory micro-environment; restriction and/or apoptosis of activated stellate cells; degradation of ECM; reduction of oxidative stress balance, and hepatocytes regeneration. In case the liver in CLD becomes chronic, the final result is cirrhosis, the potential development of hepatocarcinoma (HCC), and acute-on-chronic liver failure [28,29,30]. It is worth mentioning that in the absence of another treatment, the only established therapy for CLD patients is a liver transplant [29]. The liver transplant, although a routine procedure, is limited due to donor shortages and donor organ risks, such as cancer, infection, and autoimmune disease, generating a problem of quality and quantity in liver transplantations [31]. Based on the above, we propose that MaR1 could be an interesting pharmacological agent for the treatment and reversion of CLD.

According to previous studies, omega-3 fatty acids may exert protective effects in CLD. Petinelli et al. describe that in NAFLD, patients showed a marked increase in n-6 PUFA/n-3 PUFA ratio (that promotes steatosis). Omega-3 fatty acids may promote nuclear receptor modulation, such as inhibiting the sterol regulatory element-binding protein 1 (SREBP1c) and activating the peroxisome proliferator-activated receptor alpha (PPARα) [32,33,34]. Moreover, Enguita et al. have shown that DHA is deficient not only in plasma, but also in livers of cirrhotic patients, and that a decrease in DHA levels is correlated with the progression of the disease [35]. Interestingly, DHA also contributes to protecting the gut microbiota and intestinal wall integrity [36,37]. The administration of DHA in different liver fibrosis models showed that DHA affects the cytosolic sequestration of NF-κB subunits, explaining the capability of this omega-3 fatty acid to down-regulate inflammation, fibrosis, and oxidative stress and therefore liver damage [38,39]. NF-κB is a key master regulator of inflammation [40], and it has been widely demonstrated that EPA + DHA are down-regulators of NF-κB in acute or chronic liver disease [35,41,42,43]. Since DHA is the precursor of various SPM, including MaR1, the results observed in our model are consistent with a hepatoprotective role of MaR1.

In our study, MaR1 normalized the biochemical parameters of liver function, accompanied by a reduction in the hepatomegaly observed in the fibrosis model. This might relate to the antioxidant and anti-inflammatory effects promoted by MaR1. GSH is particularly concentrated in mammal liver tissue and is oxidized to GSSG. The GSH:GSSG ratio is often used as a marker of cellular toxicity [44]. GSH is not only a direct scavenging of reactive oxygen species, but also has roles in the network of survival, regulating necrosis and apoptosis, as well as cell signaling, and it can indirectly modulate NF-κB pathways [45]. It is worth mentioning that the GSH production is regulated by Nrf2, and at the same time Nrf2 is a potential target to modify liver fibrosis [45,46]. As GSH has a role in several redox sensitive transcription factors, such as NF-κB and activator protein-1 (AP-1), both regulate the expression of iNOS. Faced to a GSH reduction, iNOS expression is induced in liver tissue [47]. The role of iNOS in liver fibrosis has been studied using iNOS knockout mice and specific iNOS inhibitors, resulting in reduced liver fibrosis [48]. Some studies have demonstrated that MaR1 suppresses iNOS levels [13,49,50]; however, here we show for first time that MaR1 suppresses iNOS in a chronic model of liver disease. MaR1 promotes an anti-inflammatory response by an enhancement of IL-10 with a concomitant decrease in proinflammatory cytokines, TNF-α and IL-1β. This MaR1 anti-inflammatory effect has been established elsewhere [51,52,53]; in particular, the capability to enhance IL-10 levels has been demonstrated in liver injury [54]. The anti-inflammatory and pro-resolutive activity of MaR1 could be mediated by their capacity to shift macrophage to an M2 anti-inflammatory phenotype [8], where MaR1 incubation with macrophages resolution, by increasing phagocytosis and efferocytosis [55], and an epoxy-DHA (eMaR), stimulates the conversion of the M1 macrophage to M2 phenotype [56]. Macrophage polarization is inseparable from the process of resolving inflammation [57]. Han et al. have recently described that MaR1 enhances the mRNA levels of the RAR-related orphan receptor *alpha* (RORα), with induction of M2 macrophages with augmented expression of Klf4, Arg1, and Cd163 intrinsic markers of M2 phenotype [58]. Moreover, the authors in this model describe the hepatoprotective effect of MaR1 on a chronic metabolic liver disorder (NASH). Our model (DEN-induced hepatic damage) focuses specifically on the fibrotic process, after chemical cytotoxic damage has been induced. Therefore, our data establishes a role of MaR1 in preventing hepatic fibrosis development, independently of liver metabolic disorders. This highlights the potential use of MaR1 in advanced stages of chronic liver disease.

In CLD, continuous ECM remodeling during liver injury leads to an altered and excessive ECM proteoglycans accumulation, thus worsening fibrosis, which is responsible for the morbidity and mortality associated with liver failure [59]. The ECM deposit serves as an anchor for cytokines, chemokines, and growth factors which, in turn, modulate the immunity system. TGF-β1 is stored in the ECM and responds to the perturbations in the microenvironment to ensure ECM homeostasis. TGF-β1 release in response to changes in ECM rigidity during fibrosis can drive both pro-inflammatory and inhibitory immune responses [60,61]. In relation to MaR1, animals protected with MaR1 presented less ECM accumulation and fibrotic areas. Sun et al. (pulmonary fibrosis), Han et al. (NASH), and Tang et al. (mesangial cell) showed that MaR1 ameliorates pulmonary fibrosis by inhibiting TGF-β1, cell migration, fibroblast differentiation and collagen expression [58,62,63]. Moreover, we showed that MaR1 normalized the levels of TGF-β1 and its receptor, explaining the anti-fibrotic effect observed in our model. Along with ECM-normalized deposition, MaR1 was shown to induce a regenerative and anti-apoptotic phenotype. Physiological apoptosis allows the removal of dying cells without the release of proinflammatory cytokines, and a minimal immune response. However, in pathophysiological situations, the balance between cell proliferation and cell death is often altered, with the consequent loss of tissue homeostasis becoming the onset of several liver diseases [64]. Persistent apoptosis is a feature of chronic liver disease; fibrogenesis is stimulated by constant hepatocyte apoptosis, resulting in cirrhosis of the liver and a loss of hepatocytes in CLD [65]. Liver apoptosis has a dual response, depending on the cell type analyzed. For example, stellated cell death is a mechanism important for the removal of activated myofibroblast, and resolution of hepatic fibrosis [66]. This is explained by trans-differentiation of hepatic stellate cells into myofibroblast, the main cellular source of ECM and the major driver of liver fibrogenesis, comprised on 15% of all parenchymal tissue [67]. In the liver, hepatocyte apoptosis occurs mainly via two pathways: mitochondria-mediated (intrinsic) or death receptor-mediated (extrinsic) [64]. The extrinsic pathway is originated by a cytokine receptor family named the death receptor, which includes the TNF-receptor I (TNF-R1), Fas/CD95 and tumor necrosis factor related apoptosis-inducing ligand receptors 1 and 2 (TRAIL-R1 and TRAIL-R2)) by their cognate ligands (TNF-α, Fas ligand (FasL)/CD95L, TRAIL) [68]. Moreover, NF-κB, c-Jun, and p53 mediate apoptosis through up or down-regulation of apoptosis-related gene expression in the nucleus [69]. Apoptosis is indeed a proinflammatory process in liver pathophysiological conditions; the engulfment of apoptotic bodies by Kupffer cells enhances the expression of death ligands, especially the Fas ligand and TNF-α, accelerating hepatocyte apoptosis and eliciting hepatic inflammation [70]. MaR1 restores the enhanced levels of TNF-α and normalizes the apoptosis evidenced in liver fibrosis. Antiapoptotic results were accompanied by an increased level of cyclin D1 and Ki67, in a clear intention to promote survival and activate mitotic cascades in hepatocytes. Previously, Serhan et al. have demonstrated that MaR1 administration is able to regulate inflammatory resolution and tissue reparation, due to its capacity to promote tissue regeneration in planaria [71]. After that, *Wang et al.* showed that MaR1 promotes re-epithelization, accelerating wound healing in a model of tooth extraction [72]. Moreover, we previously described that MaR1 induced hepatocyte proliferation in an acute model of liver damage [54]. The molecular mechanism of the MaR1-induced repair has not yet been dilucidated. It has recently been shown that MaR1 induced hypertrophy in neonatal cardiomyocytes in culture, through a RORα/IGF-1/PI3K/Akt pathway [73].

IGF-1 and IGF-1R are molecules that play a relevant role in proliferation and apoptosis regulation (inhibiting programmed cell death), growth and cellular development through autocrine and paracrine mechanisms [74,75]. IGF is predominantly secreted by hepatocytes, and in the liver regulates mitochondrial functions, oxidative stress, and inflammation among other protective effects [76,77,78,79]. Yao et al., described in a meta-analysis that IGF is down-regulated in NAFLD patients [80]. Moreover, recombinant IGF-1 or the transfer of IGF from human umbilical cells ameliorates liver fibrosis [81,82], decreasing α-SMA and Masson’s trichomic stain, EMC. The authors showed that the IGF protective effect was related to a notorious decrease in TGF-β1 signaling. The IGF signaling pathways include a mitochondrial transcription program related to the BCL-2 induction and the mitochondria-protective signal that was coordinated through the cytoprotective transcription factor Nrf2 [83]. The dysregulation of Nrf2 pathways in the vascular system in IGF-1 deficient mice promotes the endothelial dysfunction associated with increased apoptosis [84]. Nrf2 activation is mediated via PI3K-Akt; however, Nrf2 can also be activated by an Akt/IGF-1 dependent pathway. It has been described that the activation of Nrf2 antagonizes the inflammatory pathways of TGF-β1 and NF-κB, and plays a cytoprotective role in cell damage [23,85].

Furthermore, the ROS increase mediated by TGF-β1 represses Nrf2 signaling [86], and it could be hypothesized that the Keap1–NRF2 system participates in the inhibition of TGF-β1 signaling [86,87]. It is therefore noteworthy that Nrf2 and NF-κB cross-talk among them, where the absence of the first can exacerbate cytokine production, whereas NF-κB can modulate Nrf2 transcription and activity [88]. Then, the relationship between IGF-1, Keap1 and Nrf2 observed in our study can be explained by their capability to decrease ROS/inflammation and their related molecules, by the modulation of the TGF-β1/NF-κB pathway.

In summary, MaR1 can exert hepatoprotective effects by means of an IGF-1/Keap1/Nrf2 enhanced activity, generating an anti-fibrotic, anti-inflammatory, antioxidant and tissue restoration/regeneration environment. 

## Figures and Tables

**Figure 1 cells-10-03406-f001:**
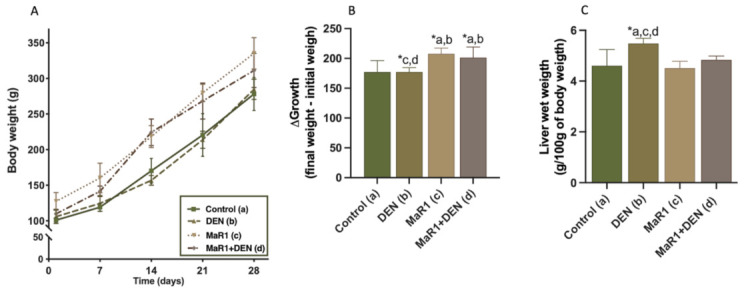
Body and liver weight of the four groups of rats. (**A**) body growth curve of rats. Body mass (g) as a function of time (weeks) of treatment; (**B**) variation of the final and initial mass after treatment with DEN and/or MaR1; (**C**) wet liver mass in relation to body mass (g/100 g). Data are expressed as mean ± SD, N = 12 animals per experimental group. One-factor ANOVA test, Tukey’s post-test, * *p* <0.05 compared among the groups. a = control; b = DEN; c = MaR1 and d = MaR1 + DEN.

**Figure 2 cells-10-03406-f002:**
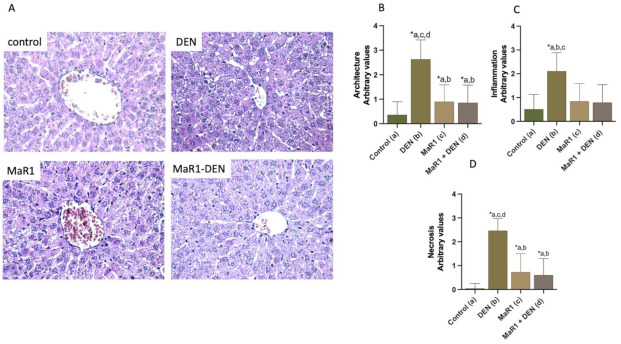
Effect of MaR1 on liver morphology and quantification of histopathological parameters observed by HE staining. (**A**) Representative liver sections stained with hematoxylin–eosin. Scores of livers sections were graphed for (**B**) architecture, (**C**) inflammation, (**D**) necrosis. At least 20 fields for every sample were analyzed at 400× magnification. n = 12 animals per experimental group. Significance was assessed by one-way ANOVA and the Tukey’s post-test. Asterisk (*) indicates *p* < 0.05. The letters identify the experiments that are compared and present this statistical difference. a = control; b = DEN; c = Mar1 and d = MaR1 + DEN.

**Figure 3 cells-10-03406-f003:**
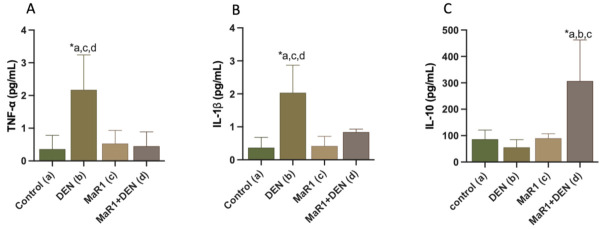
Effect of MaR1 on inflammatory mediators. Serum levels of inflammatory cytokines (**A**) tumor necrosis factor (TNF)-α; (**B**) interleukin (IL)-6 and (**C**) anti-inflammatory cytokine IL-10 levels were quantified. N = 8–12 rats per experiment. Significance was assessed by one-way ANOVA and the Tukey’s post-test. Asterisk (*) indicates *p* < 0.05. The letters identify the experiments that were compared and present this statistical difference. a = control; b = DEN; c = Mar1 and d = MaR1 + DEN.

**Figure 4 cells-10-03406-f004:**
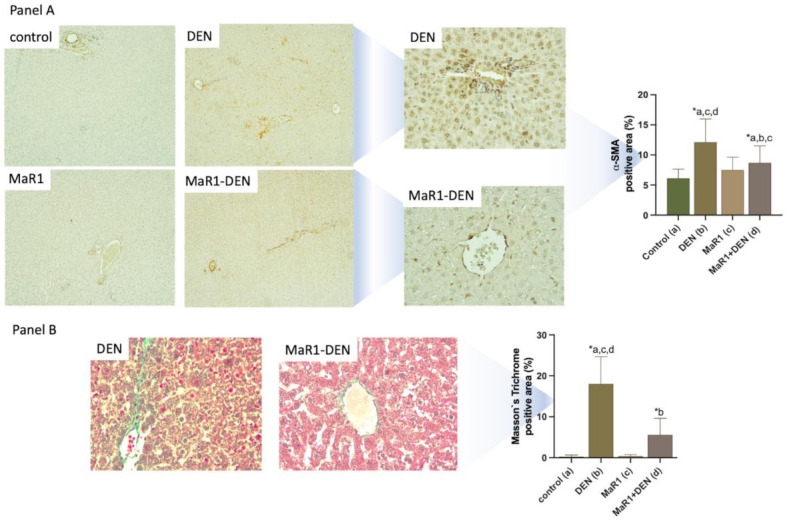
Effect of MaR1 treatment on extracellular matrix deposition**. Panel A:** representative photomicrographs of liver sections with α-SMA staining and quantification of the percentage of α-SMA positive areas; **Panel B:** representative photomicrographs of liver sections with Masson’s Trichrome staining and quantification of the percentage of positive areas. At least 20 fields were analyzed with 100× and 400× magnification per sample, from each experimental group. n = 8–12 rats per experiment. Significance was assessed by one-way ANOVA and Tukey’s post-test. Asterisk indicates *p* < 0.05. The letters identify the experiments that were compared and present this statistical difference. a = control; b = DEN; c = Mar1 and d = MaR1 + DEN.

**Figure 5 cells-10-03406-f005:**
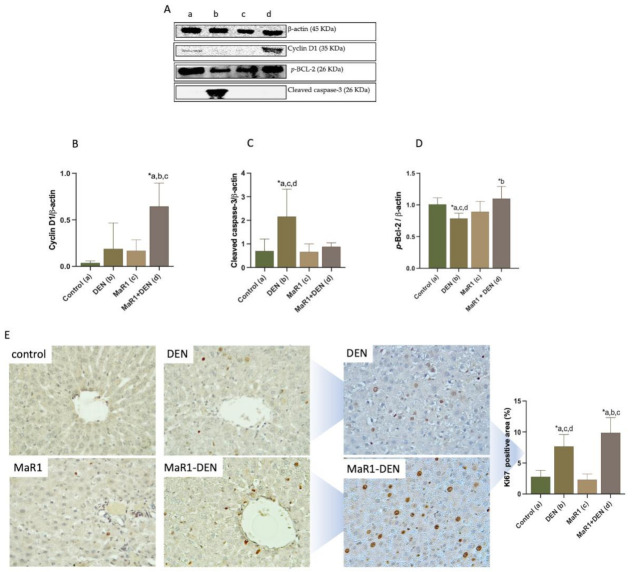
Effect of MaR1 treatment on cell cycle, proliferation and apoptosis. (**A**) Representative images of Western blots for cyclin D1, Bcl-2 and cleaved caspase-3 from liver extracts. (**B**–**D**)**,** quantification of the proteins observed showed in A. The values were normalized to β-actin; and (**E**) representative histopathological microphotography of Ki67 at 400× magnification and their quantification (Ki67 positive cell). At least 20 fields were analyzed for each sample with 100× and 400× magnification. Data are expressed as mean ± SD, n = 6–8 animals per experimental group. Significance was assessed by one-way ANOVA and the Tukey’s post-test. Asterisk indicates *p* < 0.05. The letters identify the experiments that were compared and present this statistical difference. a = control; b = DEN; c = Mar1 and d = MaR1 + DEN.

**Figure 6 cells-10-03406-f006:**
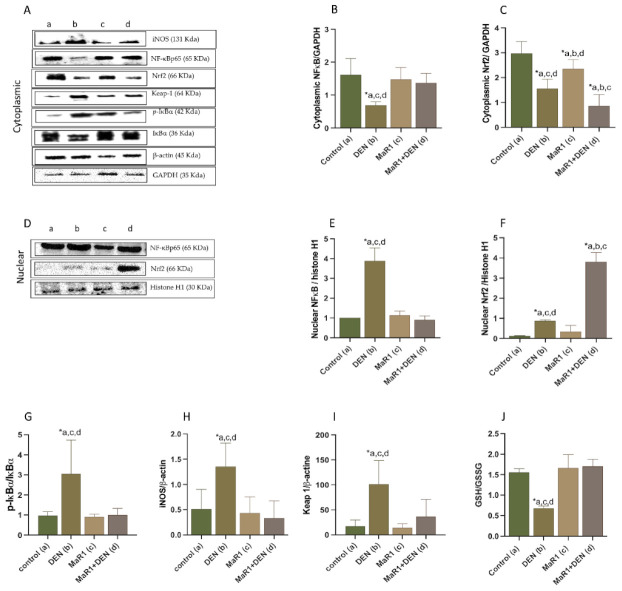
Effect of MaR1 on the activation of NF-κB and Nrf2 nuclear factors. Representative images of cytoplasmic (**A**) and nuclear (**B**) Western blots of NF-κB, Nrf2, p-IκBα, IκBα, iNOS, Keap1. Western blot analysis of (**B**) cytoplasmic and (**C**) nuclear NF-κB; (**D**) cytoplasmic (**E**,**F**) nuclear Nrf2; (**G**) p-IκBα/IκBα ratio; (**H**) iNOS; (**I**) Keap-1. (**J**) liver tissue analysis of the GSH/GSSG content. Representative images are shown in each case. The cytoplasmic levels were normalized to GAPDH or β-actin housekeeping and nuclear levels were normalized to histone H1 as housekeeping. N = 6–9 rats per experimental group. One-way ANOVA and the Tukey’s post-test. Asterisk indicates *p* < 0.05. The letters identify the groups that were compared and show the statistical difference. a = control; b = DEN; c = Mar1 and d = MaR1 + DEN.

**Figure 7 cells-10-03406-f007:**
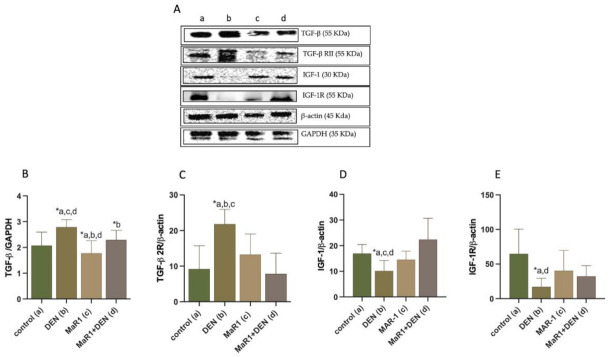
Impact of MaR1 on the levels of growth factors. (**A**) representative images of growth factors by Western Blot. Western blot analysis of (**B**) TGF-β; (**C**) TGF-β RII; (**D**) IGF-I; (**E**) IGF-IR. Representative plots are shown below each graph. TGF-β and IGF-1 levels were normalized to GAPDH or β-actin. Asterisk indicates *p* < 0.05, one-way ANOVA and the Tukey’s post-test. N= 6–9 rats per experimental group. The letters identify the groups that were compared and present the statistical difference. a = control; b = DEN; c = Mar1 and d = MaR1 + DEN.

**Table 1 cells-10-03406-t001:** Biochemical parameters of liver damage. Serum clinical values of the Sprague-Dawley rats. Data are expressed as mean ± SD. The * means *p* < 0.005.

Parameters		Grupos				
	Control	DEN		MaR1	MaR1 + DEN	
Albumin g/dL	2.39 ± 0.57	1.61 ± 0.18 *	(a,c,d)	2.53 ± 0.59	2.07 ± 0.42	
AST UI/L	90.87 ± 30.68	199.7 ± 54.94 *	(a,c,d)	85.49 ± 29.54	107.0 ± 28.32 *	(a,b,c)
ALT UI/L	46.7 ± 23.57	89.58 ± 40.63 *	(a,c,d)	55.47 ± 33.63	50.35 ± 23.46	
γGT UI/L	1.63 ± 0.52	2.2 ± 0.45		1.67 ± 0.52	1.43 ± 0.53	
AP UI/L	591.3 ± 138.6	917.8 ± 285 *	(a,c)	568.5 ± 130.5	747.0 ± 168.4	
LDH UI/L	332.7 ± 202.8	620.6 ± 239.3		531.6 ± 441.3	599.4 ± 420.6	
Hepatic index	4.610 ± 0.63	5.480 ± 0.21 *	(a,b,c)	4.515 ± 0.27	4.84 ± 0.152	

AST: aspartate transaminase; ALT: Alanine transaminase; γGT: gamma glutamil-transaminase; AP: alcaline phosphatase; LDH: lactate deshidrogenase.

## Data Availability

All data is available on request at MDPI.

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
