# Peer review of "Maresin-1 Prevents Liver Fibrosis by Targeting Nrf2 and NF-κB, Reducing Oxidative Stress and Inflammation"

_cells, 2021, doi:10.3390/cells10123406_

Round 1

Reviewer 1 Report

MaR1 is known to protect several liver diseases, such as acute liver injury, liver inflammation, NAFLD and the progressive NASH, among which MaR1 is also known to decrease NASH-associated liver fibrosis by acting as an endogenous ligand of RORa through MaR1/ROR/12-LOX autoregulatory circuit (PMID: 30855276).  This study further described an anti-fibrosis role of MaR1 in DEN-induced liver fibrosis. The phenotype is marked while the mechanism is mostly descriptive. The results are of some meaning to further support the beneficial role of MaR-1 in combating liver fibrosis. 

  1. How choosing DEN as the fibrosis model further benefits the advance of this research field needs to be described when there is already one published study of NASH fibrosis (PMID: 30855276). For example, is the current study could conclude new information such as MaR-1 inhibits liver fibrosis independent of its anti-obesity role comparing the the former studies related to this topic?
  2. Western blot data are of poor quality. An expert in western blot figure making is suggested to be involved to help improve this. The source data of uncut western blots need to be provided.
  3. The writing of this manuscript needs to be further improved to more focus on the current finding and topic, especially for the introduction part and discussion. More background information introduction could be provided to show the role of MaR-1 in treating liver diseases via which mechanism, while introduction for how CLD develops needs to be concise. 
  4. The quality of figures needs to be further improved for both western blot and histology. For example, the background color of some histology data such as Figure 5 needs attention. It is yellow. White balance could be adjusted during taking pictures.  And statistical significance needs to be marked for Panel B in Figure 5. 
  5. The analyses are descriptive and organization could be improved.  For examples, in Figure 4, although it showed 6 panels, however, Figure A-D are same meaning. GSH data could be organized with NRF2 data, while iNOS data could be organized with other inflammation-related data. qPCR of NRF2 target genes and qPCR of hepatic proinflammatory genes are needed. 
  6. Nomination of proteins and genes needs attention all through the manuscript. Here just list some examples, NrF2 needs to be NRF2 for protein description. keap-1 needs to be KEAP1. 
  7. The mechanism is lacking. For example, could MaR-1 decrease DEN-induced hepatocyte damage in vitro (inflammation, oxidative stress, macrophage polarization, etc) that the authors found in DEN-treated mice in vivo?  
  8. The potential link between current study with the former study could be discussed in more details and if possible to have data to support. For examples, did MaR1 activate ROR in this DEN-induced liver fibrosis model? 
  9. The language could be further improved. For example, in Introduction, "removal dying cells from macrophages" could be "removing...". 

Author Response

Comments and suggestions

Reply

How choosing DEN as the fibrosis model further benefits the advance of this research field needs to be described when there is already one published study of NASH fibrosis (PMID: 30855276). For example, is the current study could conclude new information such as MaR-1 inhibits liver fibrosis independent of its anti-obesity role comparing the the former studies related to this topic?

The work by Han et al (PMID: 30855276) is very interesting research about the role of MaR1 on chronic liver injury, but they use a metabolic/high fat model.

In our study we are analyzing the fibrosis (DEN model) independent of metabolic liver disorders. DEN model is a very good model for understanding the progression to advanced fibrosis-cirrhosis and HCC.  So, we can infer that MaR1 hepatoprotective effect in our model is independent of liver damage related to metabolic disorders.

We added a phrase related to this at the final of the third paragraph of the discussion.

Western blot data are of poor quality. An expert in western blot figure making is suggested to be involved to help improve this. The source data of uncut western blots need to be provided.

The images were improved for better visualization. Also, the original blots were sent to the editor and are available for all who want to see them.

The writing of this manuscript needs to be further improved to more focus on the current finding and topic, especially for the introduction part and discussion. More background information introduction could be provided to show the role of MaR-1 in treating liver diseases via which mechanism, while introduction for how CLD develops needs to be concise. 

Thanks for the suggestion.

We have added at the end of the third paragraph (introduction) a phrase related to the hepatoprotective actions of MaR1.

The quality of figures needs to be further improved for both western blot and histology. For example, the background color of some histology data such as Figure 5 needs attention. It is yellow. White balance could be adjusted during taking pictures.  And statistical significance needs to be marked for Panel B in Figure 5. 

The images were improved and the yellow background was adjusted (deleted).

The statistical significance of Figure 4 (ex-fig 5) was added.

The analyses are descriptive and organization could be improved.  For examples, in Figure 4, although it showed 6 panels, however, Figure A-D are same meaning. GSH data could be organized with NRF2 data, while iNOS data could be organized with other inflammation-related data. qPCR of NRF2 target genes and qPCR of hepatic proinflammatory genes are needed. 

We appreciate this comment very much.

The figures were re-organized according to the reviewer's request. The ex-fig 4 was deleted and the information was placed in the new-fig 6 (iNOS with NF-kB and GSH/GSSG with Nrf2). Also, we created a supplementary 6 to describe all the other assays of GSH and GSSG analysis.

Related to qPCR assays, we mentioned this problem to the editor: since we need a couple of months to ensure accurate results in our real-time PCR machine (due to pandemic situation and need for standardization)

Nomination of proteins and genes needs attention all through the manuscript. Here just list some examples, NrF2 needs to be NRF2 for protein description. keap-1 needs to be KEAP1. 

NRF2 corresponds to the acronyms of nuclear respiratory factor-2 (NRF2) instead of Nrf2 (nuclear factor erythroid 2-related factor 2). The last one was studied in our research

In the case of Keap1 (kelch like ECH associated protein 1) is common to write in capital letters when is referred to human gen. (we verified in protein bank and for Rodentia is whited as “Keap1”.

The mechanism is lacking. For example, could MaR-1 decrease DEN-induced hepatocyte damage in vitro (inflammation, oxidative stress, macrophage polarization, etc) that the authors found in DEN-treated mice in vivo? 

This is an excellent point. Mechanistically, although experimental evidence clearly indicates that macrophages are a direct target for Mar1 in liver disease, we do not discard a direct effect on hepatocytes. This is based on the observation that Mar1 application activates the hepatic antioxidant pathway response. Since this effect is difficult to explain based only on macrophages polarization switch, we speculate that Mar1 also acts on hepatic cells. Unfortunately, currently, we are unable to perform this sort of in vitro experiment.

The potential link between current study with the former study could be discussed in more details and if possible to have data to support. For examples, did MaR1 activate ROR in this DEN-induced liver fibrosis model? 

The RORa is a very interesting mechanism that can explain at the nuclear level the positive effects of MaR1 in CLD. We are planning new studies to ensure this molecular relation.

The language could be further improved. For example, in Introduction, "removal dying cells from macrophages" could be "removing...". 

Thanks to the reviewer for the recommendation. We have revised English grammar thoroughly.

Reviewer 2 Report

In the manuscript entitled "Maresin-1 prevents liver fibrosis by targeting NrF2 and NF-κB, reducing oxidative stress and inflammation", the Authors showed the strong inhibitory effect of maresin 1 – a derivative of ω-3 docosahexaenoic acid (DHA), on the liver fibrosis progression. In this study the Authors indicated the profibrotic changes in liver tissue of the Sprague-Dawley rats by injections of diethylnitrosamine, which resulted in the enhanced expression of fibrotic markers (α-smooth muscle actin, α-SMA). This effect was significantly attenuated by the maresin 1. The Authots postulated that the effect of maresin-1 on the liver fibrosis based on the attenuation of the NF-κB nuclear accumulation with concomitantly enhanced levels of Nrf2 nuclear fraction. Moreover, this effect was confirmed by the analyzes of pro-fibrotic and anti-fibrotic cytokines: TNF-α, IL-1β and IL-10. This manuscript is interesting, well described, results are novel and within the scope of the journal. However, there are a few issues that should be addressed before publication:

Comment 1: Is the specific reason, why the authors presented all data as means ± SD but not as means ± SEM?

Comment 2: It is not clear what test were used to check the normal distribution before the using ANOVA statistical test for performed analysis. Usually, in this type of studies and with such a small number of probes/repetities within the experimental / control group, it is difficult to obtain a normal distribution.

Comment 3: In the section 2.8 Statistical analysis, the Authors wrote that two-way ANOVA were used to assess the statistical significance of differences between groups, whilst in the following sections in manuscript the one-way ANOVA were is written. Can the Authors decide and standardize the version?

Comment 4: The quality of some Western blots should be improved (Fig. 4E, 6A, 7A-C, F; 8A, D). In my opinion, the Authors applied too long time of membranes exposition. Due to this fact, the densitometric analyses may give a signal inconsistent with reality. Can the Authors relate to it?

Comment 5: Have the authors investigated the effect of maresine 1 on PPARα levels and functions in this model or plan such research?

Comment 6: In my opinion graph B on the figure 7 is duplicate in D – the same axis titles as well as values on the chart. Western membranes indicate different relations in the Nrf2 to GAPDH. Can Authors paste the correct chart?

Comment 7: In the section Results (page 10/18) in the sentence: …”Next, we analyzed growth factors (Fig. 8G-J)…” – is the making error (in the figure 8 are only A – D graphs).   

Comment 8: In the figure 6D - in the axis title should be added 'p-' symbol before Bcl-2 - if it is a quantification of p-Bcl-2 from 6A picture to GAPDH?

Author Response

Comments and suggestions

Reply

·        

Comment 1: Is the specific reason, why the authors presented all data as means ± SD but not as means ± SEM?

·        

Standard deviation (SD) quantifies dispersion (how much the values vary from one another) of our dataset relative to the mean. The description of the variation among the samples (observations) and lets out to know the uncertainly and confidence of our data.  Also, we follow the premise

If data are normally distributed, the sample should be described using the mean (SD)”  and the SEM could underestimate the variability within the sample

Current recommendations from scientific communities encourage or emphasize rigor (see some web links), where the most accepted statistical description for our type of analysis is the use of mean plus SD.

https://journals.plos.org/plosone/article?id=10.1371/journal.pone.0110364

https://www.nature.com/articles/nmeth.2659

https://journals.asm.org/doi/fuOlsemll/10.1128/IAI.71.12.6689-6692.2003

https://www.graphpad.com/guides/prism/latest/statistics/statwhentoplotsdvssem.htm

Comment 2: It is not clear what test were used to check the normal distribution before the using ANOVA statistical test for performed analysis. Usually, in this type of studies and with such a small number of probes/repetities within the experimental / control group, it is difficult to obtain a normal distribution.

·        

For normal distribution, we follow the Prism recommendations (https://www.graphpad.com/support/faq/testing-data-for-normal-distrbution/) (as Kolmogorov-Smirnov the most pre-test realized).

Also, all the assay shown has analyzed by unpaired T-student previous to ANOVA (or other post-test)

*We rephrased the “statistical analysis” section for a better description.

Comment 3: In the section 2.8 Statistical analysis, the Authors wrote that two-way ANOVA were used to assess the statistical significance of differences between groups, whilst in the following sections in manuscript the one-way ANOVA were is written. Can the Authors decide and standardize the version?

·        

Thanks, we corrected the mistake and standardized. The analysis was made by one-way ANOVA.

Comment 4: The quality of some Western blots should be improved (Fig. 4E, 6A, 7A-C, F; 8A, D). In my opinion, the Authors applied too long time of membranes exposition. Due to this fact, the densitometric analyses may give a signal inconsistent with reality. Can the Authors relate to it?

·        

We have improved most images for the Western blots shown (the representative ones). The original blots were sent to the editor.

With respect to the densitometric signal, we have some problems with the last luminol kit, because in some sensible antibodies such as cleaved caspase and IkBa the signal appears at second 1 and if we decrease the antibody concentration, the signal disappears.

Comment 5: Have the authors investigated the effect of maresine 1 on PPARα levels and functions in this model or plan such research?

The reviewer is correct on this point.

We are starting to assay PPARa and SBREP in our laboratory, but we do not have enough information for publishing yet.

Comment 6: In my opinion graph B on the figure 7 is duplicate in D – the same axis titles as well as values on the chart. Western membranes indicate different relations in the Nrf2 to GAPDH. Can Authors paste the correct chart?

Thanks to the reviewer. The chart was corrected.

Comment 7: In the section Results (page 10/18) in the sentence: …”Next, we analyzed growth factors (Fig. 8G-J)…” – is the making error (in the figure 8 are only A – D graphs).

Thanks to the reviewer. The phrase was corrected

Comment 8: In the figure 6D - in the axis title should be added 'p-' symbol before Bcl-2 - if it is a quantification of p-Bcl-2 from 6A picture to GAPDH?

Thanks to the reviewer The “p” related to phosphorylation was added to the chart.

Round 2

Reviewer 1 Report

The manuscript is improved to some extent following the comments. Some new data, especially the cultured cellular experiments for in vitro studies data as well as some other minor experiments, were not performed as requested. 

Author Response

We really appreciated your review and opinion about enriching the comprehension of MaR1 mechanism in liver fibrosis through a new experimental setting (in vitro assays) but at this moment in our laboratory, we do not have the infrastructure and time enough to ensure accurate results. We understand that the work with in vitro models would be a very important set of analyses, but for the moment it is not possible to perform them.